# Antibacterial Activity of Defensive Secretions from the Lace Bug *Stephanitis svensoni* (Drake) (Hemiptera: Tingidae)

**DOI:** 10.3390/insects15040257

**Published:** 2024-04-08

**Authors:** Nobuhiro Shimizu, Chihiro Takahara, Hiroki Ogami

**Affiliations:** Faculty of Bioenvironmental Science, Kyoto University of Advanced Science, 1-1 Nanjo Otani, Sogabe, Kameoka 621-8555, Kyoto, Japan; chihiro.t.work@gmail.com (C.T.); fern.farm.pharm@gmail.com (H.O.)

**Keywords:** lace bug, *Stephanitis svensoni*, antibacterial activity, defensive secretion, GC–MS analysis

## Abstract

**Simple Summary:**

All developmental stages of *Stephanitis svensoni* (Drake) (Hemiptera: Tingidae) are found on the underside of Japanese star anise *Illicium anisatum* leaves, and nymphs of the species possess glandular setae on their dorsal abdomen, from which droplets are secreted. Secretions from nymphs of the genera *Stephanitis* and *Corythucha*, belonging to the family Tingidae, are suggested to function as defensive substances against predators in a previous study. In this study, secretions from nymphs of *S*. *svensoni* were demonstrated to contain 11 compounds, including aliphatic aldehydes, aliphatic ketones, and aromatic polyketides. Antibacterial activity examination of the 10 identifiable compounds using the paper disk method showed that four compounds exhibited antibacterial activity against the Gram-positive bacterium *Staphylococcus aureus*, while two compounds exhibited antibacterial activity against the Gram-negative bacterium *Escherichia coli*. Of the two compounds that showed antibacterial activity against both bacteria, one compound showed significant antibacterial activity even at low concentrations, indicating a stronger and wider antibacterial spectrum.

**Abstract:**

Nymphs of *Stephanitis svensoni* (Drake) (Hemiptera: Tingidae) have numerous glandular setae on their dorsal abdomens. Chemical analysis of the exudates from these setae revealed the presence of 11 compounds, including aliphatic aldehydes, aliphatic ketones, and aromatic polyketides. Among them, 3-oxododecanal, 5-hydroxy-2-heptylchromanone, and 5-hydroxy-2-undecanylchromanone were identified for the first time in the family Tingidae. Previous research has suggested that secretions from nymphs of the genus *Stephanitis*, belonging to the family Tingidae, function as defensive substances against predators. The exudates of *S. svensoni* showed antibacterial activity against the Gram-positive bacterium *Staphylococcus aureus*. Antibacterial tests conducted using preparations of the 10 identified compounds showed antibacterial activity in 3-oxododecanal, 2,6-dihydroxyacetophenone, and 1-(2,6-dihydroxyphenyl)dodecan-1-one. In addition, antibacterial tests against the Gram-negative bacterium *Escherichia coli* showed activity in 2,6-dihydroxyacetophenone and 1-(2,6-dihydroxyphenyl)dodecan-1-one. Therefore, 2,6-dihydroxyacetophenone and 1-(2,6-dihydroxyphenyl)dodecan-1-one exhibited a wide antibacterial spectrum. Particularly, 1-(2,6-dihydroxyphenyl)dodecan-1-one, which showed antibacterial activity even at low concentrations, holds promise as lead drug compound.

## 1. Introduction

Approximately 300 genera and 2500 species of the order Hemiptera, family Tingidae, are known worldwide [1]. Many among them are called “lace bugs” because of the delicate, lace-like appearance of their hemelytra and pronotum. With a few exceptions, these species are phytophagous and display host-specificity. Both adults and nymphs usually inhabit the underside of leaves, employing their exceedingly slender stylets to pierce the epidermis and extract plant nutrients. This feeding behavior causes severe damage to leaves, exerting adverse effects on plants of agricultural and horticultural importance [2]. The Japanese star anise *Illicium anisatum* (“Shikimi” in Japanese) is a well-known evergreen tree used in Buddhist ceremonies and as an ornamental plant in parks. It is widely distributed in the central and southern parts of the Japanese archipelago [3]. The neurotoxin anisatin has been isolated from Shikimi seeds [4]. *Stephanitis svensoni* (Drake) (Hemiptera: Tingidae), which feeds on Shikimi, is very similar to the andromeda lace bug *S. takeyai* (Drake et Maa), also of the same genus but larger, with an adult body length of ~5 mm. *S. takeyai* and the azalea lace bug *S. pyrioides* (Scott) originated in Japan and subsequently invaded North America [2].

Nymphs of many species of the family Tingidae possess numerous glandular setae on their dorsal abdomen, from which droplets are secreted [5]. Unique polyketides have been identified in these secretions from genera *Stephanitis* and *Corythucha*, distinguishing them from other species [6,7,8,9,10]. The biological function of these compounds remains unclear, but they are hypothesized to serve as defensive substances against predators [10,11,12,13,14]. In our repellent test, ants exhibit repellent behavior and wiping of the antennae when the secretions from nymphs of *S*. *svensoni* are brought close to their antennae, suggesting that the secretions have a certain defensive effect.

Furthermore, these compounds exhibit various properties such as antibacterial and antifungal activity, inhibition of nematode growth [15], inhibition of prostaglandin H synthase [16], and repellence of birds [17]. Chemical ecology research on the family Tingidae has focused on two invasive species of the genus *Corythucha* (*C. ciliate* (Say) and *C. marmorata* (Uhler)), leading to the identification of alarm pheromones [18,19] and sex pheromone [20].

Herein, we conducted chemical analyses to elucidate the antibacterial activity of the secretions from nymphs of *S. svensoni*. Aliphatic aldehydes, aliphatic ketones, and aromatic polyketides were identified, and the antibacterial activity of the secretions was qualitatively evaluated using the paper disk method.

## 2. Materials and Methods

### 2.1. Insects

We collected *S. svensoni* specimens feeding on the underside of leaves of *I. anisatum* plants on the Kyoto University of Advanced Science campus. The collected specimens were transported to the laboratory and raised at room temperature (~25 °C), being regularly supplied with fresh Shikimi leaves for sustenance. The leaves were replaced three times per week.

### 2.2. General Procedures

All the chemicals and reagents used were of reagent grade. Hexane, ethyl acetate, methanol, ethanol, acetone, 2-undecanone, decanal, dodecanal, magnesium sulfate, agar, anhydrous toluene, sodium metal, ethyl formate, sodium sulfate, 1,3-cyclohexanedione, pyridine, diethyl ether, acetonitrile, hydrochloric acid, sulfuric acid, triethylamine, acetic acid, mercury (II) acetate, sodium acetate, sodium hydrogen carbonate, potassium carbonate, phosphomolybdic acid hydrate, *p*-anisaldehyde, and celite were purchased from Wako Pure Chemical Industries, Ltd. (Osaka, Japan). Nonacosane, lauric anhydride, 2,6-dihydroxyacetophenone, octanoyl chloride, and palladium-activated carbon (Pd 10%) were purchased from Tokyo Chemical Industry Co., Ltd. (Tokyo, Japan). Dichloromethane, tryptone, and yeast extract were purchased from Nacalai Tesque (Kyoto, Japan). Chloroform-d (contains 0.05% (*v*/*v*) tetramethylsilane (TMS)) and neutral alumina were purchased from Sigma-Aldrich (Tokyo, Japan). Column chromatography was performed using a Wakosil silica gel C-200 with specified solvents. Thin-layer chromatography (TLC) analyses were performed on TLC silica gel 60 F254 plates (Merck, Darmstadt, Germany). The spots were visualized by UV irradiation (254 nm) and by spraying with the following reagents: phosphomolybdic acid (5 g phosphomolybdic acid and 100 mL ethanol) and *p*-anisaldehyde (1.9 mL *p*-anisaldehyde, 2.5 mL sulfuric acid, 1.2 mL acetic acid, and 68 mL ethanol). After spraying the reagents, the TLC plates were heated to 120 °C for 3 min until spots were visualized. They were characterized by a retention factor (Rf). Rf was calculated as the ratio of the travel distance of the spot to the travel distance of the solvent front. ^1^H- and ^13^C-nuclear magnetic resonance (NMR) spectra were recorded on a Biospin AC400M spectrometer (400 MHz for ^1^H and 100 MHz for ^13^C, Bruker, Yokohama, Japan) with TMS as the internal standard. Chemical shifts (δ) are given in parts per million (ppm), and coupling constants are given in Hz. The multiplicity of ^1^H-NMR signals is reported as follows: s = singlet, d = doublet, t = triplet, m = multiplet, br = broad. Gas chromatography–mass spectrometry (GC–MS) analysis was conducted using a Network GC System (6890N; Agilent Technologies Inc., Tokyo, Japan) coupled with a mass selective detector (5975 Inert XL; Agilent Technologies Inc.) operating at 70 eV. An HP-5MS capillary column (0.25 mm i.d. × 30 m, 0.25 μm film thickness; Agilent Technologies Inc.) was used with helium as the carrier gas at a flow rate of 1.00 mL/min, employing a split-less mode and a temperature program ranging from 60 °C (2 min holding time) to 290 °C at a rate of 10 °C/min, maintained at 290 °C for 5 min.

### 2.3. Preparation of the Defensive Extract

The extraction method used in this study was as follows: A 100 μL glass insert was inserted into a 1.5 mL GC glass vial, and five fourth- or fifth-instar nymphs were placed into the glass insert using a brush. Defensive secretions were extracted with hexane (10 μL) for 3 min. The resultant extract (1 μL) was injected into the GC–MS instrument using a 10 μL microsyringe. For antibacterial assays, 10 fourth- or fifth-instar nymphs were immersed in hexane (30 μL) for 3 min. The extract was then applied to a paper disk (8 mm diameter, thin, Toyo Roshi, Tokyo, Japan), and the solvent was air-dried on a glass plate for at least 1 h.

### 2.4. Quantitative Analysis of the Defensive Extract

The hexane extracts of secretions from *S. svensoni* nymphs underwent independent quantitative analysis using GC–MS. A 100 μL glass insert was placed into a 1.5 mL GC glass vial, and one insect was introduced into the glass insert using a brush. The secretions were extracted by adding 10 μL of a hexane solution containing tetradecane (5 ng) as an internal standard. After 3 min, 1 μL of the extract was injected into the GC–MS instrument using a 10 μL microsyringe. This analysis was repeated five times for third- or fifth-instar nymph extracts.

### 2.5. Antibacterial Activity Test

For the antibacterial activity test, *Staphylococcus aureus* (NBRC 12732) and *Escherichia coli* (NBRC 3972) were employed, and the antibacterial activity of the test compounds was qualitatively evaluated using the paper disk method. *S. aureus* and *E. coli* were used as glycerol stocks. Two types of bacteria were inoculated onto their respective agar media (composition: tryptone (5 g), yeast extract (1 g), MgSO_4_ (0.5 g), agar (7.5 g), and distilled water (500 mL)) and cultured overnight (cultured at 37 °C for *S. aureus* and 30 °C for *E. coli*). Colonies from the agar medium specific to each bacterium were collected using a platinum loop and suspended in a liquid medium (same composition as the agar medium except for the agar). *S. aureus* was preincubated at 37 °C for 12 h, while *E. coli* was preincubated at 30 °C for 12 h. Next, these culture solutions were mixed with agar medium and dispensed into sterile Petri dishes (90 mm in diameter × 15 mm in height). A paper disk (8 mm diameter) containing only the test compound and a solvent (hexane or methanol) as a control was placed in a Petri dish. *S. aureus* was cultured at 37 °C, while *E. coli* was cultured at 30 °C for one day.

### 2.6. Synthesis

#### 2.6.1. 3-Oxododecanal (**4**)

3-Oxododecanal (**4**) was prepared according to the literature [21]. First, 1.0 g (5.88 mmol) of 2-undecanone was dissolved in anhydrous toluene (10 mL) while stirring in ice. Then, 203 mg (8.82 mmol) of sodium metal was added while stirring the mixture for 30 min. Subsequently, 653 mg (8.82 mmol) of ethyl formate was added dropwise while maintaining the temperature below 5 °C. The mixture was stirred at that temperature for 2 h and then stirred overnight at room temperature. To cease the reaction, 20 mL of water was added to the reaction solution and stirred for 30 min. The reaction solution was extracted twice with hexane (10 mL), after which the aqueous layer was made acidic by adding 2*N* HCl. The aqueous layer was extracted three times with EtOAc (10 mL) and then the organic layer was washed with water and brine. The organic layer was dried over anhydrous Na_2_SO_4_ and then evaporated *in vacuo*. The resulting oil was purified using SiO_2_ column chromatography (hexane:EtOAc = 5:1) to yield aldehyde **4** (211 mg, 18%).

MS (EI) *m*/*z* (rel. %): 198 (M^+^, <1), 180 (2), 155 (3), 124 (6), 99 (14), 86 (76), 71 (100), 55 (15), 43 (30).

#### 2.6.2. 1-(2,6-Dihydroxyphenyl)dodecan-1-one (**8**)

1-(2,6-Dihydroxyphenyl)dodecan-1-one (**8**) was prepared according to the literature [22]. We dissolved 1,3-cyclohexanedione (0.56 g, 5.0 mmol), lauric anhydride (1.91 g, 5.0 mmol), and pyridine (0.44 g, 5.5 mmol) in CH_2_Cl_2_ (5 mL) and allowed the reaction to proceed at room temperature for 30 min. After the reaction, the solvent was concentrated under reduced pressure, and the mixture was partitioned between hexane/diethyl ether (1:1) and chilled 1*N* HCl. The organic layer was dried over anhydrous Na_2_SO_4_, followed by evaporation of the solvent *in vacuo* to yield 3-oxocyclohex-1-en-1-yl dodecanoate. The crude product was dissolved in CH_3_CN (15 mL), after which Et_3_N (1 mL) and acetone cyanohydrin (40 µL) were added. The resulting mixture was stirred overnight at room temperature. After evaporation of the reaction mixture *in vacuo*, the residue was partitioned between hexane/diethyl ether (1:1) and water. After washing of the organic layer with 1*N* HCl and water, it was dried over anhydrous Na_2_SO_4_. The solution was evaporated *in vacuo* to obtain 2-dodecanoyl-3-hydroxycyclohex-2-en-1-one. This crude product was dissolved in AcOH (5 mL), and 1.26 g of Hg(OAc)_2_ (4.0 mmol) and 0.33 g of NaOAc (4.0 mmol) were added. The mixture was then stirred at 120–125 °C until the precipitate dissolved. The reaction solution was cooled to room temperature, 1*N* HCl was added, and the mixture was stirred for 30 min. The mixture was filtered through cerite and washed with hexane/EtOAc (3:1), and the organic layer was then washed with water, saturated NaHCO_3_ aq., and brine. Subsequently, the organic layer was dried over anhydrous Na_2_SO_4_, filtered through neutral alumina, and washed with EtOAc. After evaporation of the solution *in vacuo*, it was purified using SiO_2_ column chromatography (hexane:EtOAc = 5:1) to afford ketone **8** as a pale yellow solid (0.91 g, 62%).

^1^H-NMR (400 MHz, CDCl_3_) δ 9.60 (2H, brs), 7.21 (1H, t, *J* = 10.0 Hz), 6.39 (2H, d, *J* = 8.0 Hz), 3.12 (2H, t, *J* = 6.0 Hz), 1.69 (2H, m), 1.30 (16H, m), 0.88 (3H, t, *J* = 6.0 Hz). ^13^C-NMR (100 MHz, CDCl_3_) δ 208.02 (CO), 161.17 (C= ×2), 135.65 (C=), 110.07 (C=), 108.44 (C= ×2), 44.83 (CH_2_), 31.92 (CH_2_), 29.65 (CH_2_), 29.57 (CH_2_ ×2), 29.42 (CH_2_ ×2), 29.36 (CH_2_), 24.44 (CH_2_), 22.70 (CH_2_), 14.13 (CH_3_). MS (EI) *m*/*z* (rel. %): 292 (M^+^, 7), 274 (7), 189 (11), 165 (25), 152 (23), 137 (100), 81 (6), 41 (6).

#### 2.6.3. Synthesis of the Standard 2-alkyl-5-hydroxychromanone

Chromanone (**6**) was prepared according to the literature [10,23,24]. We dissolved 1 g of 2,6-dihydroxyacetophenone (6.57 mmol) and 4.96 g of K_2_CO_3_ (35.9 mmol) in acetone (33 mL). After reacting at room temperature for 15 min, 1.12 mL of octanoyl chloride (6.57 mmol) was added dropwise, and the mixture was refluxed for 24 h. Afterward, water (30 mL) was added, and 1*N* HCl was introduced until the pH of the reaction solution reached ~4. Extraction was then conducted with EtOAc. The organic layer was washed with brine, dried over anhydrous Na_2_SO_4_, and the solvent was evaporated *in vacuo*. The crude product was purified using SiO_2_ column chromatography (hexane:EtOAc, 20:1) to yield 2-heptyl-5-hydroxychromone (598 mg). Subsequently, 100 mg (0.38 mmol) of the product was dissolved in EtOAc (10 mL), to which 10% Pd/C (100 mg, 0.094 mmol) was added. The mixture was vigorously stirred at room temperature under a hydrogen atmosphere (1 atm) for 3 h. The reaction solution was filtered through celite and washed with EtOAc. The solvent was evaporated *in vacuo* to afford chromanone **6** (73 mg, 73%). Chromanones **7** and **9** were synthesized using the same method, employing decanoyl chloride or dodecanoyl chloride, respectively.

#### 2.6.4. 2-Heptyl-5-hydroxychroman-4-one (**6**)

^1^H-NMR (400 MHz, CDCl_3_) δ 11.7 (1H, s, OH), 7.36 (1H, t, *J* = 8.3 Hz), 6.45 (1H, d, *J* = 8.3 Hz), 6.43 (1H, d, *J* = 8.3 Hz), 4.44 (1H, m), 2.71 (2H, m), 1.68 (2H, m), 1.47 (10H, m), 0.85 (3H, t, *J* = 5.8 Hz). ^13^C-NMR (100 MHz, CDCl_3_) δ 198.75 (CO), 162.05 (C=), 161.74 (C=), 138.16 (CH=), 109.06 (CH=), 108.21 (C=), 107.34 (CH=), 77.48 (CH), 42.26 (CH_2_), 34.81 (CH_2_), 31.76 (CH_2_), 29.32 (CH_2_), 29.15 (CH_2_), 24.84 (CH_2_), 22.64 (CH_2_), 14.16 (CH_3_). MS (EI) *m*/*z* (rel. %): 262 (M^+^, 48), 163 (79), 152 (4), 137 (100), 108 (16), 81 (6).

#### 2.6.5. 2-Nonyl-5-hydroxychroman-4-one (**7**)

^1^H-NMR (400 MHz, CDCl_3_) δ 11.7 (1H, s, OH), 7.34 (1H, t, *J* = 8.3 Hz), 6.48 (1H, d, *J* = 8.3 Hz), 6.42 (1H, d, *J* = 8.3 Hz), 4.40 (1H, m), 2.70 (2H, m), 1.68 (2H, m), 1.47 (14H, m), 0.88 (3H, t, *J* = 5.8 Hz). ^13^C-NMR (100 MHz, CDCl_3_) δ 198.75 (CO), 162.05 (C=), 161.73 (C=), 138.15 (CH=), 109.06 (CH=), 108.21 (C=), 107.34 (CH=), 77.23 (CH), 42.26 (CH_2_), 34.82 (CH_2_), 31.89 (CH_2_), 29.52 (CH_2_), 29.49 (CH_2_), 29.36 (CH_2_), 29.30 (CH_2_), 24.84 (CH_2_), 22.69 (CH_2_), 14.13 (CH_3_). MS (EI) *m*/*z* (rel. %): 290 (M^+^, 51), 163 (97), 137 (100), 108 (16), 41 (7).

#### 2.6.6. 2-Undecanyl-5-hydroxychroman-4-one (**9**)

^1^H-NMR (400 MHz, CDCl_3_) δ 11.7 (1H, s, OH), 7.34 (1H, t, *J* = 8.3 Hz), 6.48 (1H, d, *J* = 8.3 Hz), 6.42 (1H, d, *J* = 8.3 Hz), 4.41 (1H, m), 2.71 (2H, m), 1.69 (2H, m), 1.47 (18H, m), 0.88 (3H, t, *J* = 5.8 Hz). ^13^C-NMR (100 MHz, CDCl_3_) δ 198.75 (CO), 162.06 (C=), 161.74 (C=), 138.15 (CH=), 109.06 (CH=), 108.21 (C=), 107.34 (CH=), 77.35 (CH), 42.27 (CH_2_), 34.82 (CH_2_), 31.92 (CH_2_), 29.63 (CH_2_), 29.59 (CH_2_), 29.55 (CH_2_), 29.49 (CH_2_), 29.45 (CH_2_), 29.35 (CH_2_), 24.84 (CH_2_), 22.70 (CH_2_), 14.15 (CH_3_). MS (EI) *m*/*z* (rel. %): 318 (M^+^, 61), 282 (3), 163 (100), 137 (92), 108 (17), 41 (13).

## 3. Results and Discussion

We conducted chemical analyses to clarify the antibacterial activities of *S. svensoni* nymph secretions. Extraction of nymphs with hexane followed by gas chromatography–mass spectrometry (GC–MS) revealed the presence of the various compounds in nymph secretions (Figure 1): decanal (**1**); 2-undecanone (**2**); dodecanal (**3**); 3-oxododecanal (houttuynin) (**4**); 2,6-dihydroxyacetophenone (**5**); 5-hydroxy-2-alkylchromanones (**6, 7,** and **9**); 1-(2,6-dihydroxyphenyl)doecan-1-one (**8**); and nonacosane (**11**). Details regarding GC retention times and MS spectra of these natural products are summarized in Table 1, while compound structures are summarized in Figure 2. When hexane extracts of the leaves from the host plant, *I. anisatum*, were analyzed by GC–MS, no compounds identical to those of the nymph extracts were detected, except for hydrocarbons. Furthermore, no previous papers on the components of *I*. *anisatum* leaves have described compounds detected in nymph extracts [25,26,27,28]. Thus, compounds **1**–**10** were suggested to be exocrine secretions from nymphs.

For compounds **1**, **2**, **3**, **5**, and **11**, analysis of commercially available samples via GC–MS confirmed their structures through spectral data matching with natural products. Compound **4**, chromanones **6**, **7**, and **9**, and aromatic ketone **8** were determined through MS spectrum searches using the Agilent NIST05 Mass Spectral Library alongside synthesized compounds exhibiting high match rates [10,21,22,23,24]. Subsequently, the compound structures were determined by matching their GC–MS spectral data with those of the natural products. However, compound **10**, presumed to be a chromanone based on its MS spectrum, awaits structural determination. Notably, GC–MS analysis of adult insects yielded only high-boiling-point hydrocarbons, with compounds **1**–**10** absent.

Compounds **1** and **2** were previously detected as nymph exudates of *S*. *pyrioides* [6]. Compound **3** is newly reported, but aldehydes with different carbon chain lengths have been detected in *S*. *pyrioides* [6]. Compound **4**, an essential oil component of *Houttuynia cordata* (Thunb.), possesses antibacterial, antiviral, and anti-inflammatory properties [29]. Sodium houttuyfonate, which is an adduct of compound **4** and sodium bisulfite, has been used clinically in China as an antibacterial agent. Dihydroxyacetophenone **5** and 5-hydroxy-2-alkylchromanone **7** were identified as nymph secretions in *S*. *pyrioides* and *S*. *takeyai* [6,10]. Analogously, 5-hydroxy-2-alkylchromanones **6** and **9,** featuring different carbon chain lengths, serve as structural analogs of chromones in nymph secretions of the rhododendron lace bug *S*. *rhododendri* (Horváth) [7]. Dihydroxyaromatic ketone **8** has been identified from *S*. *takeyai* [10], suggesting these chromones and dihydroxyaromatic ketones are characteristic for genus *Stephanitis* in the family Tingidae. The biological roles of these secondary metabolites remain incompletely understood, but they are hypothesized to function as defensive substances against predators [10,11,12,13,14]. For example, red-winged blackbirds, such as *Agelaius phoeniceus*, actively avoid the nymph secretions of *S*. *pyrioides* [17].

Herein, we examined the biological activity of *S*. *svensoni* nymph secretions by investigating their antibacterial activity against the Gram-positive bacterium *S*. *aureus*. Antibacterial assays involved applying a hexane extract from 10 *S*. *svensoni* nymphs onto a paper disk, resulting in the formation of a 10 mm inhibition zone. To elucidate the active compounds in the crude hexane extract, qualitative antibacterial tests were conducted using samples of compounds **1**–**9** and **11**, tested at 30 and 300 µg (Table 2).

Results showed inhibition zones for compounds **4**–**6** and **8** at 300 µg, with compound **8** exhibiting antibacterial activity even at 30 µg. Similarly, antibacterial tests against the Gram-negative bacterium *E*. *coli* using the same paper disk method displayed inhibition zones for compounds **5** and **8** at 300 µg, with compound **8** retaining antibacterial activity at 30 µg. Compounds **5** and **8** showed antibacterial activity against both *S*. *aureus* and *E*. *coli*, with compound **8** demonstrating notable antibacterial activity even at low concentrations, indicating a stronger and wider antibacterial spectrum. Quantitative analysis via GC–MS indicated that 2.5 ± 0.52 and 5.8 ± 1.4 ng of compound **8** was extracted from third- and fifth-instar nymphs, respectively, suggesting a higher production in fifth-instar nymphs by ~2.3 times than in third-instar nymphs. Dihydroxyaromatic ketones, such as compounds **5** and **8**, showed antibacterial activity, but their analogs, chromanones, did not show antibacterial activity except for compound **6**.

Antibacterial and antinematode activities of nymph secretions from genera *Stephanitis* and *Corythucha* of the family Tingidae have been reported [15]. Chromanone **7** displayed antibacterial activity against two Gram-positive bacteria (*Clavibacter michiganensis* and *Rhodococcus facians*), while compound **8** exhibited antibacterial activity against *C*. *michiganensis* and growth inhibition against the parasitic nematode *Ascaris suum*. The pharmacological effects of compounds **5** and **8** were also investigated, confirming their inhibition of prostaglandin H synthase to a degree comparable to or greater than aspirin [16]. In our ongoing research, we aim to quantitatively evaluate the antibacterial activity of compound **8**, newly discovered in the present study, and elucidate its antibacterial spectrum, encompassing fungi alongside bacteria. Furthermore, we will endeavor to conduct structure-activity relationship research, seeking to elucidate the mechanisms of action and explore the potential for this compound to emerge as a leading drug candidate.

## 4. Conclusions

We extracted droplets exuded from the glandular setae present on the dorsal abdomens of nymphs of *S. svensoni* using hexane and conducted GC–MS analysis, detecting 11 compounds. Antibacterial activity examination of the 10 identifiable compounds using the paper disk method showed that compounds **4**–**6** and **8** exhibited antibacterial activity against the Gram-positive bacterium *S. aureus*, while compounds **5** and **8** demonstrated antibacterial activity against the Gram-negative bacterium *E. coli*. Both compounds **5** and **8** showed activity against both types of bacteria, with compound **8** demonstrating activity at a smaller tested amount.

## Figures and Tables

**Figure 1 insects-15-00257-f001:**
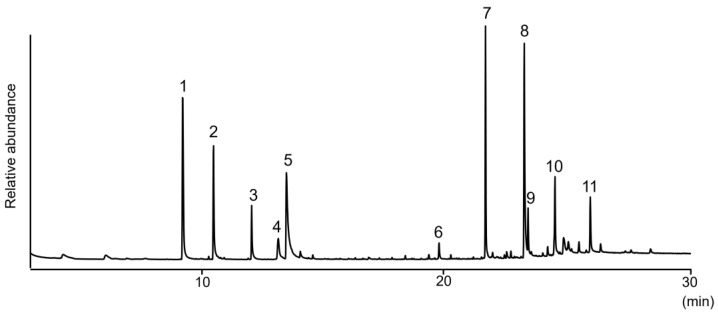
Gas chromatogram of nymph secretions from *S*. *svensoni:* (**1**) decanal; (**2**) 2-undecanone; (**3**) dodecanal; (**4**) 3-oxododecanal; (**5**) 2,6-dihydroxyacetophenone; (**6**) 5-hydroxy-2-heptylchromanone; (**7**) 5-hydroxy-2-nonylchromanone; (**8**) 1-(2,6-dihydroxyphenyl)dodecan-1-one; (**9**) 5-hydroxy-2-undecanylchromanone; (**10**) unknown; (**11**) nonacosane.

**Figure 2 insects-15-00257-f002:**
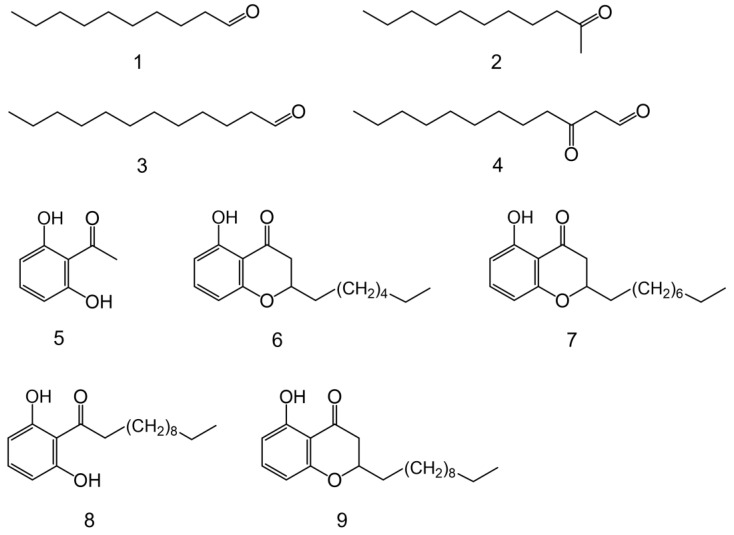
Structure of the compounds tested for antimicrobial activity: (**1**) decanal; (**2**) 2-undecanone; (**3**) dodecanal; (**4**) 3-oxododecanal; (**5**) 2,6-dihydroxyacetophenone; (**6**) 5-hydroxy-2-heptylchromanone; (**7**) 5-hydroxy-2-nonylchromanone; (**8**) 1-(2,6-dihydroxyphenyl)dodecan-1-one; (**9**) 5-hydroxy-2-undecanylchromanone.

**Table 1 insects-15-00257-t001:** GC–MS data of secretions from *S. svensoni*.

Peak	Compound	GC *t*_R_ (min)	Molecular and Diagnostic Ions *m*/*z* (Intensity %)
1	decanal	9.44	41 (100), 57(82), 70 (44), 82 (44), 95 (25), 112 (25), 128 (6), 138 (M^+^—18, 2)
2	2-undecanone	10.70	43 (97), 58 (100), 71 (37), 85 (8), 112 (6), 127 (4), 155 (3), 170 (M^+^, 7)
3	dodecanal	12.25	41 (100), 57 (93), 67 (44), 82 (63), 96 (37), 110 (16), 123 (7), 140 (14), 156 (2), 166 (M^+^—18, 1)
4	3-oxododecanal	13.33	43 (30), 55 (15), 71 (100), 86 (76), 99 (14), 124 (6), 155 (3), 180 (M^+^—18, 2)
5	2,6-dihydroxyacetophenone	13.64	43 (8), 81 (14), 108 (3), 137 (100), 152 (M^+^, 42)
6	5-hydroxy-2-heptylchromanone	19.92	81 (6), 108 (16), 137 (100), 152 (4), 163 (79), 262 (M^+^, 48)
7	5-hydroxy-2-nonylchromanone	21.83	41 (7), 108 (16), 137 (100), 163 (97), 290 (M^+^, 51)
8	1-(2,6-dihydroxyphenyl)dodecan-1-one	23.39	41 (6), 81 (6), 137 (100), 152 (23), 165 (25), 189 (11), 274 (7), 292 (M^+^, 7)
9	5-hydroxy-2-undecanylchromanone	23.57	41 (13), 108 (17), 137 (92), 163 (100), 282 (3), 318 (M^+^, 61)
10	unknown	24.67	43 (31), 108 (10), 137 (31), 162 (82), 163 (100), 177 (10), 205 (5), 332 (35)
11	nonacosane	26.16	43 (62), 57 (100), 71 (76), 85 (54), 99 (20), 113 (11), 127 (8), 267 (4), 408 (M^+^, 3)

**Table 2 insects-15-00257-t002:** Antimicrobial activity of secretions from *S*. *svensoni* against *S*. *aureus* and *E*. *coli* using the paper disk method.

Larval Secretion	*S. aureus*		*E. coli*	
	30 μg	300 μg	30 μg	300 μg
decanal (**1**)	– (2)	– (2)	– (2)	– (4)
2-undecanone (**2**)	– (2)	– (2)	– (2)	– (4)
dodecanal (**3**)	– (2)	– (2)	– (2)	– (4)
3-oxododecanal (**4**)	– (2)	11.0 (4)	– (2)	– (4)
2,6-dihydroxyacetophenone (**5**)	– (6)	18.6 (2)	– (2)	14.5 (12)
5-hydroxy-2-heptylchromanone (**6**)	– (2)	10.0 (4)	– (2)	– (4)
5-hydroxy-2-nonylchromanone (**7**)	– (2)	– (2)	– (2)	– (4)
1-(2,6-dihydroxyphenyl)dodecan-1-one (**8**)	10.6 (8)	11.7 (10)	11.8 (8)	12.2 (12)
5-hydroxy-2-undecanylchromanone (**9**)	– (2)	– (2)	– (2)	– (4)
nonacosane (**11**)	– (2)	– (2)	– (2)	– (4)

The number: the diameter (mm) of the inhibition circle. –: the inhibition circle was not recognized. The number in parentheses: the n-number.

## Data Availability

Data are included in this article.

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
