# Peer review of "Antibacterial Activity of Defensive Secretions from the Lace Bug Stephanitis svensoni (Drake) (Hemiptera: Tingidae)"

_insects, 2024, doi:10.3390/insects15040257_

Round 1

Reviewer 1 Report

Comments and Suggestions for Authors

This is an interesting study, identifying glandular compounds from lace bugs. The science is sound and the manuscript well written. I only have some minor corrections and suggestions.

Introduction:

L52: Remove "in them"

L54: "elucidate the biological activity" is too broad. You only focus on identification and evaluation of antibacterial activity.

Methods

L98 Remove "them"

2.6 Synthesis: I would like to know more details about the synthesis protocols, as for 3-oxododecanal for example, this is the first published synthetic method (!) according to SciFinder, and the protocols contain reactions that are not always in the text books (at least not in my books). It would be appropriate to add references or rational to the key steps, and I would suggest a figure with the reaction schemes to make it easier for the reader to follow.

L162: Cerite should be celite?

Results and discussion:

First sentence needs rephrasing. "Clarify the biological activities" is too vague. Specify what you did, as for the introduction.

Figure 1: Might be better to say Total Ion Chromatogram than Gas chromatogram (if that is what it is).

L251: My only scientific correction. You don't analyse what the nymphs produced, but what you were able to extract. A critical difference: It should say something along the lines of "the extracts contained" instead of "nymphs produced".

Author Response

Response to reviewer #1

We thank you for your careful reading of the manuscript and helpful comments and suggestions. We have made revisions according to your comments and suggestions, as described below.

Special comment for Introduction

  1. L52: Remove "in them"

Response

We have removed.

  1. L54: "elucidate the biological activity" is too broad. You only focus on identification and evaluation of antibacterial

Response

In the revision paper, “biological activity” has been replaced by “antibacterial activity”.

Special comment for Methods

  1. L98 Remove "them"

Response

We have removed.

  1. 6 Synthesis: I would like to know more details about the synthesis protocols, as for 3-oxododecanal for example, this is the first published synthetic method (!) according to SciFinder, and the protocols contain reactions that are not always in the text books (at least not in my books). It would be appropriate to add references or rational to the key steps, and I would suggest a figure with the reaction schemes to make it easier for the reader to follow.

Response

The papers referenced in the synthesis are indicated in L134, L150, and L179, respectively. In the process, we swapped references 23 and 24. Each synthesis method is a known, short and simple process and is therefore omitted from the figure.

  1. L162: Cerite should be celite?

Response

As you indicated, it was corrected.

Special comment for Results and discussion

  1. First sentence needs rephrasing. "Clarify the biological activities" is too vague. Specify what you did, as for the introduction.

Response

We have revised as follows: “clarify the antibacterial activities”

  1. Figure 1: Might be better to say Total Ion Chromatogram than Gas chromatogram (if that is what it is).

Response

Figure 1 is not an MS spectrum, so there is no problem in expressing it as a "gas chromatogram".

  1. L251: My only scientific correction. You don't analyze what the nymphs produced, but what you were able to extract. A critical difference: It should say something along the lines of "the extracts contained" instead of "nymphs produced".

Response

We have revised as follows (L284): “Quantitative analysis via GC–MS indicated that 2.5 ± 0.52 and 5.8 ± 1.4 ng of compound 8 was extracted from third- and fifth-instar nymphs, respectively,///”

Miscellaneous revisions

Information about reagents (L69-80), methods of TLC analysis (L82-89), and NMR spectrum notation (L91-94) were added to the “general procedures” in M&M. In addition, new references 12, 14, 15, and 25 to 29 were added as citations.

Reviewer 2 Report

Comments and Suggestions for Authors

Antibacterial Activity of Defensive Secretions from the Lace 2 Bug Stephanitis svensoni (Drake) (Hemiptera: Tingidae) is a well-written manuscript. However, there are a few questions about the conclusions drawn. The authors extracted “defensive secretions” by rinsing the insects in the solvent. How did the authors confirm that the chemicals identified as "defensive secretions" were not the chemicals from the Shikimi leaves? I would suggest authors should also perform chemical analysis of Shikimi leaves and then compare those with chemicals from the insects. At the very least, authors should do a literature review of Shikimi leaf chemicals and then compare them with insect chemicals.

Regarding antibacterial activity, are Staphylococcus aureus(NBRC 12732) and Escherichia coli (NBRC 3972) known to affect Stephanitis svensoni? If they are not known to affect S. svensoni, then characterizing the identified chemicals as "defensive secretions" seems farfetched. Identified chemicals may have antibacterial properties, but they may or may not be providing any defense to the insects. For identifying chemicals, authors carried out a literature search, but the gold standard for identifying chemicals is to follow the neat standards. I would suggest authors run the standards for the identified chemicals whenever they are available. Authors should also carry out statistical analysis of antibacterial activity.

Author Response

Response to reviewer #2

We thank you for your careful reading of the manuscript and helpful comments and suggestions. We have made revisions according to your comments and suggestions, as described below.

Special comment for defensive secretions and Shikimi leaf compounds

Antibacterial Activity of Defensive Secretions from the Lace Bug Stephanitis svensoni (Drake) (Hemiptera: Tingidae) is a well-written manuscript. However, there are a few questions about the conclusions drawn. The authors extracted “defensive secretions” by rinsing the insects in the solvent. How did the authors confirm that the chemicals identified as "defensive secretions" were not the chemicals from the Shikimi leaves? I would suggest authors should also perform chemical analysis of Shikimi leaves and then compare those with chemicals from the insects. At the very least, authors should do a literature review of Shikimi leaf chemicals and then compare them with insect chemicals.

Response

Many nymphs of the family Tingidae possess numerous glandular setae on their dorsal abdomen, from which droplets are secreted. Ants exhibit repellent behavior when the secretions of Stephanitis svensoni nymphs are brought close to their antennae, suggesting that the secretions have a certain defensive effect. This sentence was added to the introduction (L48-50). The possibility that the secretions of lace bug nymphs are defense substances is also mentioned in references 11–15 (12, 14 and 15 were newly added).

When secretions are collected using a glass capillary, dissolved in an appropriate solvent, and then subjected to GC-MS analysis, a GC chromatogram almost identical to that shown in Fig. 1 is obtained. Furthermore, when extracting Shikimi leaves with hexane, the compounds shown in Fig. 1 are not observed, nor is the formation of an inhibition zone observed in the antibacterial activity test. Four papers (26-29) were added that analyzed the essential oil components of the leaves of Shikimi leaves, but the secondary metabolites obtained from the nymphs of Stephanitis svensoni were not reported in four papers (L224-L229).

Special comment for standard compounds

Regarding antibacterial activity, are Staphylococcus aureus (NBRC 12732) and Escherichia coli (NBRC 3972) known to affect Stephanitis svensoni? If they are not known to affect S. svensoni, then characterizing the identified chemicals as "defensive secretions" seems farfetched. Identified chemicals may have antibacterial properties, but they may or may not be providing any defense to the insects. For identifying chemicals, authors carried out a literature search, but the gold standard for identifying chemicals is to follow the neat standards. I would suggest authors run the standards for the identified chemicals whenever they are available. Authors should also carry out statistical analysis of antibacterial activity.

Response

In this study, we chemically analyzed the secretions of nymphs, identified compounds using standard and synthetic compounds, and tested for antimicrobial activity using these pure compounds. In this paper, only the presence or absence of antimicrobial activity was determined for each compound using a simple paper disk method. In the future, we plan to quantitatively and statistically measure the antimicrobial activity (MIC: Minimum Inhibitory Concentration) using the samples that showed activity.

Miscellaneous revisions

Information about reagents (L69-80), methods of TLC analysis (L82-89), and NMR spectrum notation (L91-94) were added to the “general procedures” in M&M. In addition, new references 12, 14, 15, and 25 to 29 were added as citations.

Round 2

Reviewer 2 Report

Comments and Suggestions for Authors

The authors have made significant changes to the MS.